# Application of Plasma Electrolytic Oxidation Coating on Powder Metallurgy Ti-6Al-4V for Dental Implants

**Cristina Garcia-Cabezón [1],\***, **María Luz Rodriguez-Mendez [1]**, **Vicente Amigo Borrás [2]**, **Bayon Raquel [3]**, **Jose Carlos Rodriguez Cabello [4]**, **Arturo Ibañez Fonseca [4]** and **Fernando Martin-Pedrosa [1]**

[1] Materials Engineering, E.I.I., Universidad de Valladolid, Paseo del Cauce 59, 47011 Valladolid, Spain; mluz@eii.uva.es (M.L.R.-M.); fmp@eii.uva.es (F.M.-P.)
[2] Materials Technology Institute, Univesitat Politècnica de València, Camí de Vera s/n, 46022 Valencia, Spain; vamigo@mcm.upv.es
[3] Tribology Unit, Fundación IK4-Tekniker, 20600 Eibar, Spain; raquel.bayon@tekniker.es
[4] BIOFORGE Lab, University of Valladolid–CIBER-BBN, Paseo Belen 19, 47011 Valladolid, Spain; Cabello@bioforge.uva.es (J.C.R.C.); aibanez@bioforge.uva.es (A.I.F.)
\* Correspondence: crigar@eii.uva.es; Tel.: +34-98-342-3389

**Abstract:** Ti-6Al-4V alloy obtained by powder metallurgy (PM) is a good candidate biomaterial in the manufacture of dental implants but its inherent porosity makes it have worse corrosion behavior than conventionally obtained alloys. In order to improve the corrosion and biological properties, surface modification technologies could be used. The plasma electrolytic oxidation (PEO) process is a novelty process successfully used in case of conventional titanium alloys. The present work investigates the effect of PEO treatment on PM Ti-6Al-4V alloy using two electrolytes. Both coatings show good adherence and improved corrosion behavior in artificial saliva, the PEO coatings delivers a steady growth of corrosion resistance from day one until 90 days immersion. Highest corrosion resistance was shown in case of Ca/P enrichment PEO coatings. The cytocompatibility tests indicated these coatings seem to be appropriate to improve the bone osseointegration with proper porosity index.

**Keywords:** titanium based alloys; corrosion; coating; biocompatibility

---

## 1. Introduction

The Ti-6Al-4V alloy is the most widely used titanium alloy in orthopedics and dental implants [1] due to its high strength-to-weight ratio, good fatigue strength and toughness and anti-magnetic properties. Compared to cobalt alloys and stainless steel this alloy has lower elastic modulus, better corrosion resistance and biocompatibility. However, there are some problems associated with the use of such a biomaterial that are yet to be addressed.

The first one is the stress shielding problem as consequence of differences between the elastic modulus of bone and biomaterial and the associated bone resorption [2] which is one of the main causes of failure in implants [3,4]. A methodology for decreasing the elastic modulus of near-bone titanium-based alloys is the use of materials with a porous structure that can also reduce stiffness. In addition, porous materials can result in faster and more complete growth of the bone, which in return results in greater strength at the implant pores. Such increase could allow for a homogeneous transfer of stress between the bone and the implant [5–7].

Many surface modification strategies have been attempted to obtain porous titanium-based alloy materials, including PM technology [8] and oxidation [9]. In this work, a combination of both is proposed. The anodic oxidation of titanium and its alloys in different solutions is an easy and

simple method to obtain porous or tubular structures of variable thickness -in our case on material manufactured by PM.

PM materials could be one of the most suitable solutions to overcome this deficiency. These PM components have good mechanical behavior, because the porosity improves bone fixation by balancing the load transfer through the implant decreasing the effect of stress shielding. Thus, powder metallurgy (PM) processing provides a feasible and economic way of manufacturing complex shape devices with such advantages as good dimensional precision and high surface finish [10,11].

The second one is the bio-inertness of the native surface oxide of titanium alloys which results in difficulty to achieve good bone cell adhesion to the implant compared to bioactive materials such as hydroxyapatite [12]. The bioactivities of Ti and its alloys depend mainly on the structure, morphology and chemistry of its surface layers [13,14]. Therefore, appropriate surface modification techniques could lead to the improvement of their bioactivity. Coating techniques to develop biologically active coatings like ion implantation, laser surface treatment, electrodeposition [14], physical vapour deposition [15] and anodic oxidation by plasma electrolytic oxidation (PEO) have been used. This last technique has recently been successfully applied to titanium alloys [16]. The porous morphology of the titanium promotes the adhesion and proliferation of osteoblasts [17]. Furthermore, porosity also helps to promote bone ingrowth and cell attachment thereby creating a better mechanical fixation to the surrounding tissue [18,19].

PM components show a drawback related to corrosion resistance. It decreases as porosity increases due to a poor passive film stability, the increase in the area exposed to the corrosive media and to crevice corrosion susceptibility [20]. Also, one way of addressing these problems is the use of surface modification technologies, which are often utilized to improve the osseointegration ability and the corrosion resistance of titanium dental implants [21].

Over the last years, plasma electrolytic oxidation (PEO) has appeared as a breakthrough electrochemical method to produce metal-oxide-based films with customized features on the surface of valve metals. This is an electrochemical process that induces to the growth of a ceramic-like film on materials surface, generally characterized by high hardness and thickness, therefore providing wear and some corrosion resistance, as well as chemical stability, electrical and thermal insulation. PEO is an easy, rapid and versatile technique that may be applied to treat components of variable size and geometry. Also, this technology is recognized as ecologically friendly, thus becoming very appealing to be used for industrial applications [22].

The PEO process is characterized by the phenomenon of electrical discharge on the metal that acts as an anode immersed in an electrolyte at the potentials above the spark voltage of the coating [23]. The anodization occurs by applying a positive voltage that exceeds the dielectric breakdown of the oxide film. During coating growth, a short duration micro-discharge plasma is produced continuously over the coating surface, accompanied by gas evolution, which in turn promotes a porous, hard, rough, corrosion-resistant, crystalline and hydrophilic films with the incorporation of the electrolyte constituents [24]. The anodic oxide film exhibits a variety of different properties that depend on the composition and microstructure of the substrate and processing parameters, such as anode potential, electrolyte composition, temperature, and current density [25].

In recent researches the influence of substrate, electrolyte characteristics and current modes on the properties of PEO films has been studied for pure Ti [26] and Ti-6Al-4V [24,27]. The corrosion resistance is one of the most important factors analysed, showing that the improved corrosion resistance on NaCl or PBS was mainly determined by the thickness, composition and quality of the dense inner layer. On the other hand, concentration and type of electrolyte solutions also affects the corrosion resistance. For this reason, lots of researchers have paid attention to the role of several electrolytes with elements such as Ca, P, Mg, Si, Zn, and Mn on Ti and on Ti6Al4V. They showed that the modified electrolytes provided higher resistance to corrosion and better osseointegration of dental implant [28,29].

Thick, porous and well-adhered $TiO_2$ oxide films may be successfully grown in the surface of biomedical metallic implants with complex geometries, such as dental or orthopedic implants made

out of titanium- and titanium alloys [18,30,31] and other materials [32]. The electrochemical bath used for PEO processes may be enriched with bioactive elements such as calcium (Ca) and phosphorous (P), aiming their incorporation into the anodic films during electrochemical reactions that take place during the processes, accompanied by the conversion of the metal surface into titanium oxide. The addition of these elements in the electrolyte and consequent enrichment of PEO films with them, may provide not only excellent corrosion protection and enhanced tribological properties [24], but also improve the osseointegration properties of the materials.

Few studies on PM PEO titanium have been done, but they demonstrate the significant potential of surface engineered PM samples for biomedical applications, improving the chemical integration thanks to a large internal surface area covered with rough and porous oxide layers [33]. Recent studies with Ti parts made by metal injection molding [34] and by additive manufacturing [35] modified with PEO showed good microstructural and mechanical behavior. No studies have been found related to the application of these coatings to PM Ti-6Al-4V. It should also be understood that the inherent porosity of PM Ti alloys should be an advantage for the adhesion of these coatings. Furthermore, in the present work the corrosion behavior of PEO PM Ti-6Al-4V in artificial saliva (AS) will be undertaken by electrochemical techniques. In addition, corrosion resistance will be improved by modifying the PEO electrolyte solution with elements as Ca and P.

Another important advantage of PEO coatings is their ability to provide both cell adhesion and antibacterial activity. Good cellular viability results have been obtained for PEO coated Ti and Ti alloys especially when coating is composed of hydroxyapatite or calcium phosphate-based phases [36]. In this work, the effect of PEO coating, using two electrolyte solutions, on PM Ti-6Al-4V on cytotoxicity and cellular proliferation will be analyzed.

Thus, the main objective of the present work is to explore the role of PEO coatings on PM Ti-6Al-4V obtained by PM using two electrolytes solution, one of these Ca- and P-enriched. The corrosion performance in AS simulated body fluids is electrochemically evaluated by means of open circuit potential, and anodic potentiodynamic polarization curves. The correlation between microstructure and corrosion resistance is also discussed. The cellular viability of this new material is also investigated for its use for medical applications. In this work, conventional powder metallurgy materials with a low degree of porosity have been used as a first step to later work with samples of higher porosity.

## 2. Materials and Methods

### 2.1. Materials

Disc specimens of the Ti-6Al-4V alloy of 20 mm in diameter and in 6 mm of thickness were obtained via powder metallurgy using blending elemental powders. The titanium powder obtained by hydride-dehydride process, aluminum powder and vanadium powder were provided and blended by SE-Jong Materials Co. (Incheon, South Korea). Once the raw material is received, it is mixed in a 2L Inversina tube from Bioengineering AG (Wald ZH, Switzerland), for 45 min at 90 rpm in a controlled argon atmosphere. The powder particles have irregular shape; its particle size and distribution measured by a Mastersizer 2000 analyser (Malvern Panalytical, Malvern, UK) was d (0.1) = 10.5 µm; d (0.5) = 24.5 µm and d (0.9) = 49.5 µm. The powder mixture was uniaxial compacted at 600 MPa using a cylindrical floating die and were sintered at 1250 °C under vacuum conditions (<100 Pa) for 3 hours. Densities of sintered compacts were calculated with pycnometer for solids. The porosity was determined by image analysis. Table 1 show the composition, porosity and density values of the sample.

**Table 1.** Porosity, density and composition of Ti6Al4V sample.

| Open% Porosity | Closed% Porosity | Density g/cm$^3$ | Relative Density | Al% | V% | Fe% | Zr% | Ti% | O% max | N% max | H% max |
|---|---|---|---|---|---|---|---|---|---|---|---|
| 0.26 ± 0.07 | 5.42 ± 0.12 | 4.25 ± 0.09 | 94.3% ± 0.1 | 6.62 | 4.55 | 0.02 | 0.03 | 88.8 | 0.25 | 0.3 | 0.5 |

A commercial Keronite device (20 kW AC power supply, KERONITE, KT 20-50, Haverhill CB9 8PJ, UK) was used to perform PEO treatments on disc samples made of Ti-6Al-4V alloy. Before applying the PEO coating, the titanium discs were cleaned with ethyl alcohol in an ultrasonic bath and then dried with air. These samples worked as the anode, while a stainless-steel mesh incorporated into an electrochemical cell of 50 liters capacity, served as the cathode. Two different aqueous electrolytes were employed to perform PEO treatments: (1) a commercial electrolyte provided from Keronite (Keronite Electrolyte 1003-01 for titanium) which is composed of 0.1–1% sodium fluoride (NaF) and other Ca- and P-based chemical compounds, and (2) a home-made electrolyte constituted of 0.35 M of calcium acetate hydrate (Scharlab, Barcelona, Spain) and 0.02 M of β-glycerol phosphate disodium salt pentahydrate (Scharlab) as the source of Ca and P, respectively. PEO processes were carried out at 25 A/dm$^2$ for 10 min in both electrolytes, aiming to compare the influence of the electrolyte composition on the features of PEO films. After PEO process, roughness was measured using a contact roughness meter (Perthometer M2, Mahr, Nikon Instruments Inc., Melville, NY, USA).

## 2.2. Microstructural and Mechanical Characterization

Samples were polished and etched before observation by optical metallography and scanning electron microscopy with energy dispersive X-ray spectroscopy (SEM/EDS). A QUANTA 200F SEM system (FEI, Hillsboro, OR, USA) was used to record the images of the samples. Kroll' reagent etching (HNO$_3$-HF-H$_2$O) during 5 s was selected. X-ray diffraction (XRD) using an Agilent SuperNova system (Agilent Technologies XRD Products, OX5 1QU, UK) equipped with an Atlas S2 CCD (Agilent Technologies XRD Products, OX5 1QU, UK) was implemented to help identify some phases. Analysis elemental of titanium alloy and PEO-coatings were made by wavelength dispersive X-ray fluorescence (XRF) using a S8 Tiger system (Bruker, Billerica, MA, USA).

## 2.3. Corrosion Testing

Electrochemical corrosion measurements were carried out in physiological AS solution (2 g/L NaCl, 2 g/L KCl, 3.95 g/L CaCl$_2$, 1.54 g/L NaH$_2$PO$_4$, 5 g/L urea and 0.005 g/L Na$_2$S) at 37 °C ± 1. An Ag/AgCl KCl 1M electrode was used as reference electrode. Graphite was used as counter-electrode. The electrochemical methods included open circuit potential (OCP) and anodic polarization measurements.

The potentiodynamic anodic polarization curves were performed following ASTM (American Society for Testing and Materials) standard G-5 [37] using a potentiostat/galvanostat Model 273A (EG&G PAR, Gaithersburg, MD 20878, USA). Surface preparation of the samples was performed with 1 μm diamond paste polishing. The experimental test procedure was as follows: 5 min delay at open circuit potential, 2 min anodic attack at −220 mV Ag/AgCl, delay of 2 min at OCP, 1 min cathodic cleaning at −600 mV Ag/AgCl, 5 min delay at OCP and finally and anodic potentiodynamic scan, which started at 200 mV below OCP, reaching 1300 mV Ag/AgCl at 50 mV/min. Nitrogen streaming and agitation were used throughout the whole test. For each sample, it was performed an initial test and a final test corresponding to samples immersed 90 days in AS solution. Polarization was performed at least three times under each condition. The surface of the samples after the polarization was observed with an optical microscope.

The corrosion rate was determined using Tafel´s extrapolation methods. The Tafel´s slope cathodic (βc) and anodic (βa) and the corrosion current densities (i$_{corr}$) were estimated from the Tafel plots. Corrosion potential (E$_{corr}$) was also determined.

All tests were performed at least three times for each condition and there were no significant differences between each result. The coefficients of variation between these tests were less than 5% for each of the two techniques used.

### 2.4. Cytocompatibility Tests

MC3T3-E1 mouse preosteoblasts (CRL-2593™, ATCC®, Manassas, VA, USA) were cultured for expansion in complete growth medium, i.e. $\alpha$-MEM with nucleosides and without ascorbic acid (Gibco, Brooklyn, NY, USA), supplemented with 10% fetal bovine serum (Gibco Brooklyn, NY, USA), at 37 °C and 5% $CO_2$. All the cells were used at passages 3–5 for subsequent experiments.

For the cytocompatibility tests, cells were trypsinized and seeded on 24-well plates at a density of 10,000 cells/cm$^2$. They were left unaltered for attachment in complete growth medium at 37 °C and 5% $CO_2$ for 2 h. Then, the samples (untreated Ti-6Al-4V, PEO and PEO-Ca/P) were placed on the cell layer, so the material was directly in contact with the cells, as specified in the ISO 10993-5 [38]. After 1 and 3 days, cell viability (LIVE/DEAD assay; Invitrogen, Carlsbad, CA, USA) and metabolic activity (alamarBlue assay; Invitrogen, Carlsbad, CA, USA) tests were performed as recommended by the manufacturer, after removing the Ti samples from the wells. All the experiments were performed in triplicate.

Images of living and dead cells were taken with a Nikon Eclipse Ti-E coupled to a Nikon DS-2MBWc digital camera (Nikon Corporation, Tokyo, Japan) using the NIS-Elements Advanced Research software (version 4.5; Nikon Corporation, Tokyo, Japan), which allowed the automatic imaging of almost the full well surface for quantitative determination of cell viability. On the other hand, metabolic activity in terms of alamarBlue reduction was measured in a fluorescence plate reader. In both cases, a positive control, i.e. cells not in contact with samples, were used for normalization.

## 3. Results and Discussion

### 3.1. Microstructural Characterization

Figure 1a presents the general microstructure observed for the Ti-6Al-4V sample sintered in a vacuum, showing the typical duplex lamellar $\alpha + \beta$ structure with low presence of pores. Figure 1b shows the X-ray diffraction patterns of reference sample; $\alpha$-HCP (hexagonal close-packed ) is the main phase, but not the only one, a mixture of $\beta$-BCC (body centered cubic) and $\alpha$-HCP was observed. The addition of vanadium, beta stabilizer element, promote a few contribution of the $\beta$-BCC phase retained at room temperature in accordance to Figure 1a. SEM/EDS was used to identify these phases. Figure 1c shows the lamellar constituent mentioned which matrix is identified as $\alpha$-HCP phase and the $\beta$-BCC phase was retained in form of thin plates with a new phase inside them; this new phase had needle-like structure. Their result is consistent with chemical composition measured by EDS, as shown in Table 2 and Figure 1d.

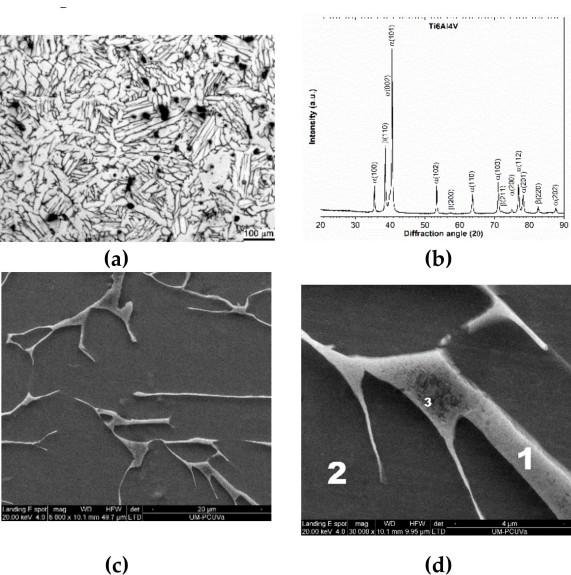

**Figure 1.** LOM micrograph (**a**) XRD diffractograms (**b**) SEM-BSE micrograph (**c**) SEM micrograph of Ti6Al4V and (**d**) SEM-EDX micrograph of Ti6Al4V (1) plate, (2) Matrix and (3) Needle.

**Table 2.** Chemical composition (wt.%) of the areas in Figure 1d measured by EDS.

| Sample | Area | Al | V | Fe | Ti |
|---|---|---|---|---|---|
| | Plate (1) | 4.11 | 15.52 | 1.15 | 79.22 |
| Ti6Al4V | Matrix (2) | 8.16 | 1.15 | - | 90.69 |
| | Needle (3) | 5.02 | 14.10 | 1.04 | 79.84 |

The β-BCC phase had lower aluminum/vanadium ratio than the matrix, α-HCP phase. The former was richer in vanadium and iron, beta stabilizer elements, and the latter was richer in aluminum, alpha stabilizer element with high solubility in solid α-HCP solution. For the thin needle-like precipitates inside β-BCC phase, only a slight increase of aluminum was observed, it is logical due the small thickness of this phase that could be identified as martensite phase.

After carrying out the PEO process with the two electrolytes, a $TiO_2$ layer with a 3D structure consisting of numerous open pores was formed in both samples (Figure 2a,b) corresponding to PEO and PEO-Ca/P samples respectively. Both samples are covered with a fine cellular-like oxide network, micropores and microcracks were observed in any of the PEO films.

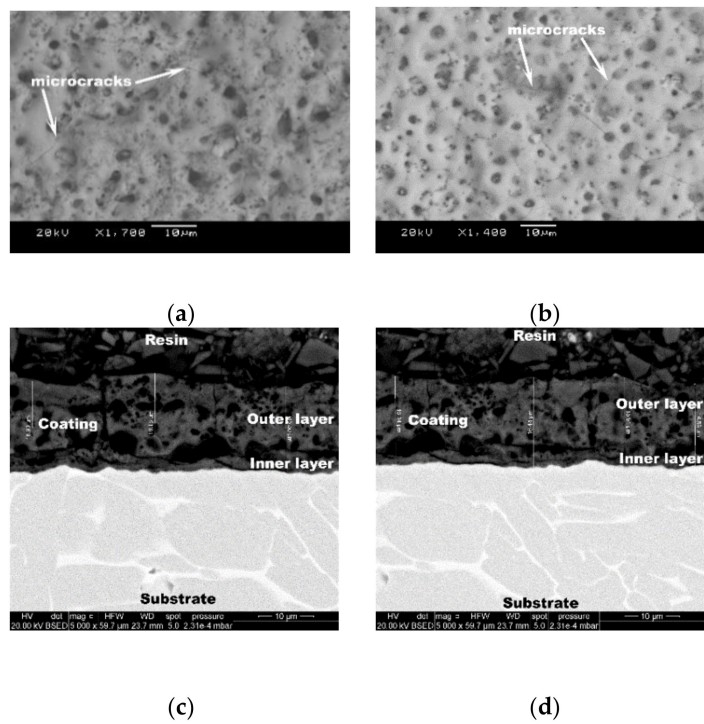

(**a**)　　　　　　　　　　　　　　　　　　(**b**)

(**c**)　　　　　　　　　　　　　　　　　　(**d**)

**Figure 2.** SEM images of the Ti6Al4V surfaces submitted to PEO process (**a**) superficial PEO sample, (**b**) superficial PEO-Ca/P sample, (**c**) cross-section PEO sample and (**d**) cross section PEO-Ca/P sample.

The backscattered electron images of the $TiO_2$ oxide layer cross section (Figure 2c,d) show similarities in thickness for both electrolytes. It seems that both the films have grown well adhered to Ti-6Al-4V substrate as it is expected to occur with films created by a PEO process. The average thickness, in μm, for PEO was 16.26 ± 1.54 and 16.02 ± 0.92 PEO-Ca/P samples. Also, the average roughness was similar both PEO samples but an important increase was observed respect to reference sample from 0.05 μm on the reference sample to 1.18 and 1.08 μm on the samples oxidized for PEO and PEO Ca/P specimens, respectively.

The biological properties of titanium dental implants depend on its surface oxide film. Measurement of the roughness parameter of the dental implant surface morphology is important because its value influences the adhesion, adsorption and differentiation of the cells [39,40]. So, one possible method of improving dental implant biocompatibility is to increase surface roughness

and decrease the contact angle. Data from the literature [39] show that the surface treatments that lead to an improvement in the biocompatibility of titanium dental implants have a medium roughness between 0.5 and 1.97 μm. In the present work, the roughness values were along this lines.

For both electrolytes, the coating is composed of two distinct layers, a loose porous outer layer with higher thickness and a compact inner layer much thinner. The compact inner layer was dense and adhered well to the substrate however an important amount of pores with connections, holes and other structure defects were present in the outer layer. This cross section surface morphology is the same that observed for non-porous materials [25,30,41]. The most important difference between the two PEO samples was detected for the EDS analysis. Table 3 shows the EDS results obtained by PEO and PEO-Ca/P samples in the coating. The results indicated that PEO sample was mainly formed by alumina and titanium oxide species although phosphorus appears to be present. No vanadium was detected; it is a good fact taking into account that high vanadium concentration has been reported as being toxic to the human body (1). When examined PEO-Ca/P sample, the layer contained more calcium and sodium than PEO layer and higher amount of elements of the substrate like aluminum and vanadium. The (Ca)/(P) ratio in this layer shows the approaching 1.48 which is near the natural bone in the case of PEO-Ca/P sample [29].

**Table 3.** EDS of PEO-treated Ti6Al4V alloy.

| Sample | Al wt% | Ca wt% | K wt% | Na wt% | P wt% | Si wt% | O wt% | V wt% | Ti wt% |
|---|---|---|---|---|---|---|---|---|---|
| PEO | 2.91 | 0.66 | 0.16 | 0.33 | 4.68 | 0.46 | 45.62 | - | 45.36 |
| PEO-Ca/P | 3.82 | 5.93 | 0.29 | 1.22 | 3.99 | 0.21 | 37.51 | 2.02% | 44.76 |

The EDS analysis in the case of PEO-Ca/P sample showed significant differences along the cross-section of the coating. Figure 3 presents the EDS mapping data for PEO-Ca/P coating showing EDS-mapping analysis images of O, Ca, P, V, Al, Si, Na and Ti elements.

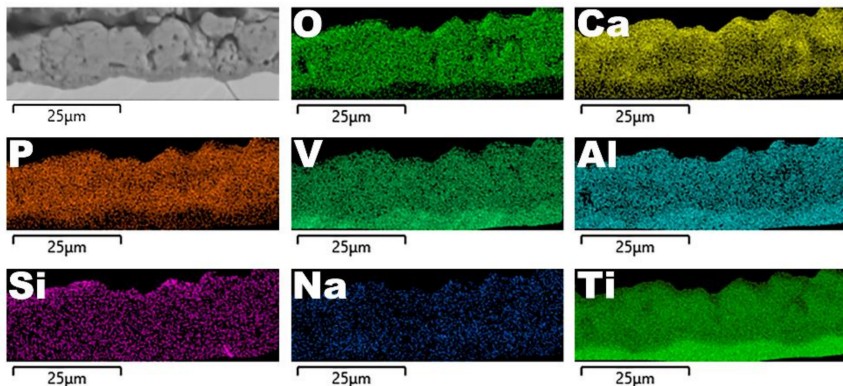

**Figure 3.** SEM and EDS mapping images of PEO/-treated Ti6Al4V alloy in electrolyte containing Ca and elements.

Oxygen is mainly detected on the coating; it is homogeneously distributed except within the pores. Elements coming from metallic substrate (V, Al and Ti) are distributed across the whole coating thickness with a higher concentration in the inner part. Ca is detected at a higher concentration on the coating surface rather than in the inner layer while P is detected at a higher concentration on the layer adjacent to substrate. Si, and Na are distributed uniformly throughout the coating.

Results of the XRF analysis of metallic elements of coating composition are shown in Table 4. It should be noticed that XRF is not an effective technique in estimating the content of light elements and hence these results have not been considered.

**Table 4.** Results of XRF analysis of coating of PEO and PEO-Ca/P samples.

| Sample | Al wt% | Ca wt% | Cr wt% | Fe wt% | K wt% | Na wt% | P wt% | Si wt% | V wt% | Ti * wt% |
|---|---|---|---|---|---|---|---|---|---|---|
| PEO | 3.56% | 0.60% | 0.26% | 0.21% | 0.11% | 1.75% | 8.00% | 1.24% | - | 84.10% |
| PEO-Ca/P | 3.64% | 7.06% | - | 0.09% | - | 0.38% | 4.60% | 0.09% | 2.43% | 81.50% |

\* Ti + light elements.

The general trend seen from the XRF results is similar for both coatings, but there were some significant differences. Elements such as Ca and V were detected in higher amounts for PEO-Ca/P coating while elements as P, Na and Fe were higher for PEO. These results are in agreement with the EDS results and confirmed the absent of vanadium in PEO sample.

Figure 4 presents the results of XRD phase composition analysis for the coated samples, which featured a superposition of sharp peaks corresponding to crystalline phases, with a broad scattering maximum located around and attributed to amorphous compounds. The coatings developed in both samples were composed of a mixture of rutile and anatase. The peak intensities for the rutile phase were higher than for the anatase phase in both coatings. Phosphate peaks were also detected for both coatings. For PEO sample a mixture of sodium titanium phosphate and titanium phosphate were identified while for PEO-Ca/P sample the intensity of calcium phosphate peaks was the highest. In addition, $\alpha$-HCP titanium peaks were visible, especially for PEO-Ca/P sample. To interpret these results, it must be considered that some $\alpha$-HCP titanium phase peaks coincide with hydroxyapatite peaks (39.5° and 40.3°)3. PEO-Ca/P sample also showed the highest intensity of amorphous calcium phosphate compounds. All these substances were observed in Ca/P enriched PEO coating for non-PM materials and was considered to promote bioactivity of Ti [31]; therefore, all of them could potentially be useful for biomedical implant applications. This section may be divided by subheadings.

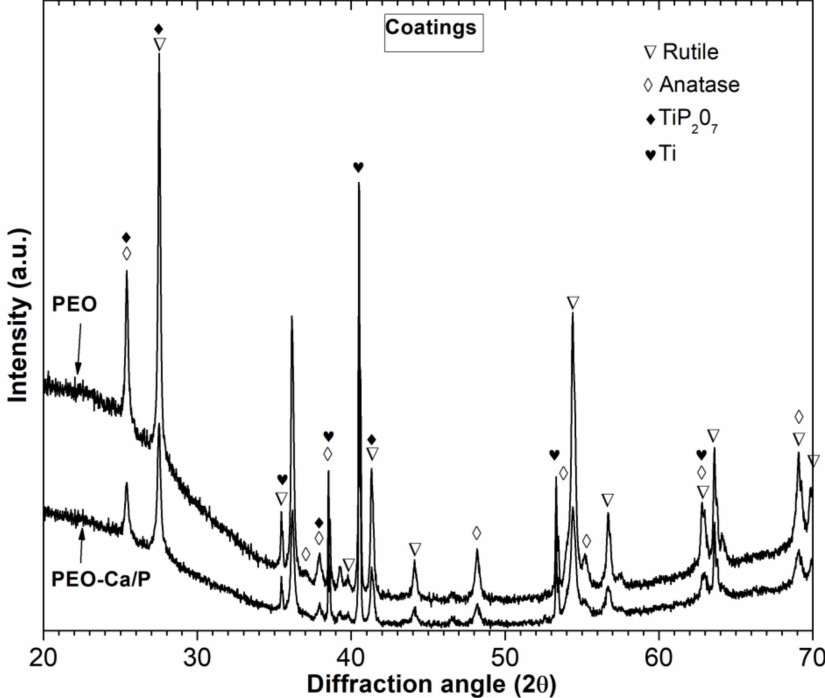

**Figure 4.** Results of XRD analysis of the coatings surfaces of PEO and PEO-Ca/P samples.

*3.2. Corrosion Behaviour*

The OCP is the potential at which the sample is in equilibrium with the environment and provides a qualitative information about the susceptibility to corrosion from a thermodynamic point of view.

Figure 5 shows the OCP evolution for every sample before and after 90 days of immersion testing in AS. The effect of PEO coatings for 0 days samples is clearly different for both coatings. The PEO-Ca/P sample showed a stable scan with nobler potential values than non-coated sample while for the PEO sample the OCP increased with the time until it stabilized at a stationary value more negative than the reference sample. A variety of parameters of the surface oxide films (composition, phases, and structural defects as porosity and cracking) could explain the different susceptibility to degradation of the modified surface. The immersion test had a significant beneficial effect for coated and non-coated samples, all samples showed, after 90 days in AS exposure, more positive potentials. The increase was especially clear for the PEO sample, so the results showed that the PEO-coated samples are electrochemically more resistant than Ti-6Al-4V and that the incorporation of Ca and P provides a more noble potential.

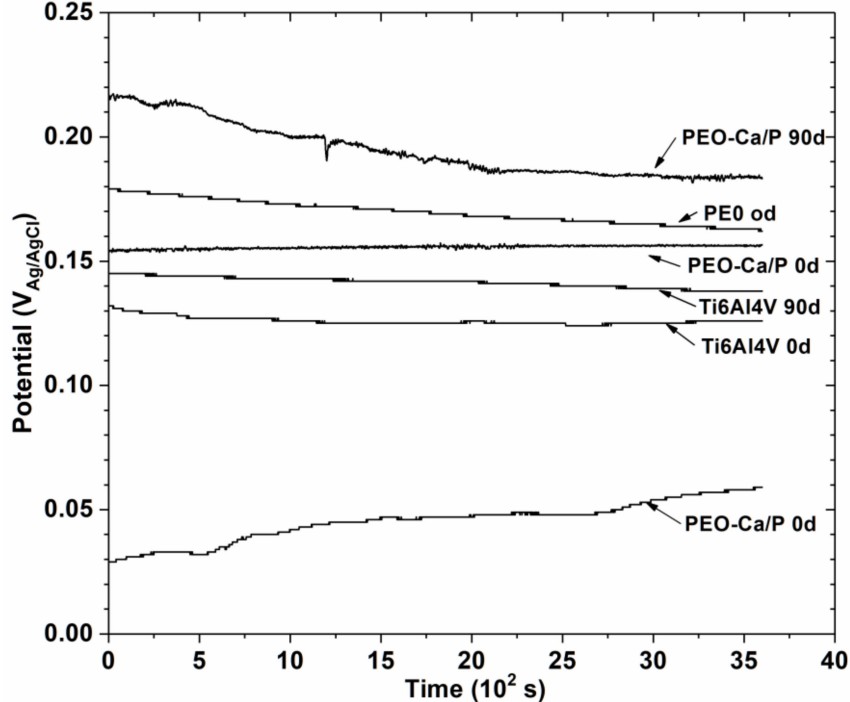

**Figure 5.** OCP evolution of Ti6Al4V, PEO and PEO-Ca/P samples in AS solution after 0 and 90 days of immersion.

The effect of PEO coating and the effect of the exposure time in AS solution upon the anodic polarization curves are shown in Figure 6. Because of the porosity of both, the substrate and the coating, the real current density cannot be correctly estimated. It is not possible to calculate such a real current density because the wet area inside the pores remains undetermined, so the current density is overestimated [42]. In Table 5, we have summarized the corrosion potential ($E_{corr}$), corrosion current density ($i_{corr}$) using Tafel extrapolation method and passive current density ($i_{pass}$) estimated from the polarization curves. The values of $i_{pass}$ were extracted from the polarization curves at the applied potential of 0.5 VAg/AgCl (potential lying within the passive region).

All curves had a similar shape: the first domain which corresponds to cathodic branch where the current density is determined by the reduction of water and partially of dissolved oxygen, and the second one that corresponds to the metal dissolution and passive layer formation. A passive layer is formed on all samples, exhibiting a broad range potential where the current density was low, without breakdown potential, even at very high potential, indicating that in the AS solution any sample for any condition does not undergo localized corrosion. However, there were some differences that will be commented.

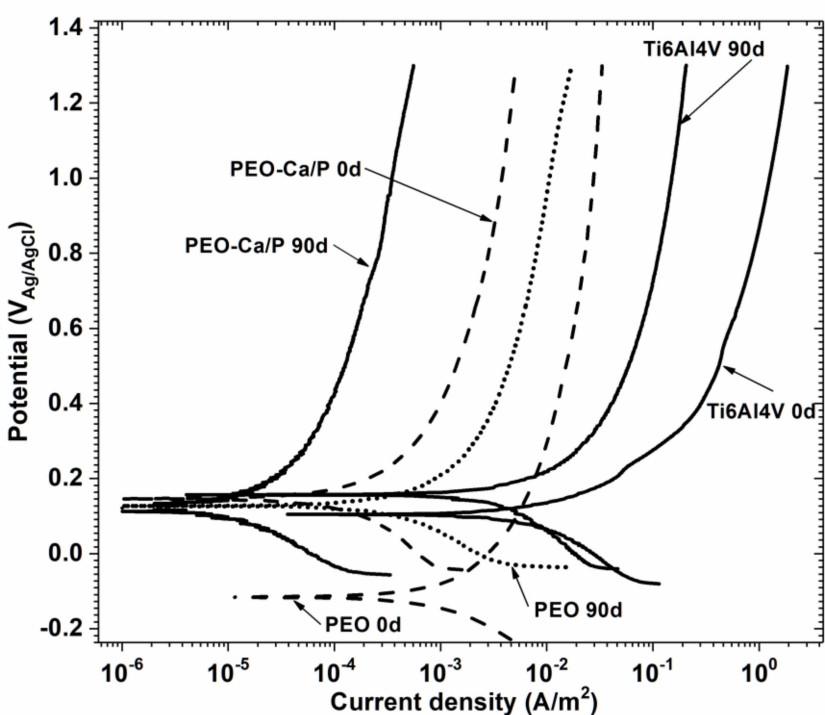

**Figure 6.** Anodic polarization curves of Ti6Al4V, PEO and PEO-Ca/P samples in AS solution after 0 and 90 days of immersion.

**Table 5.** Electrochemical parameters in AS solution at solution at 37 °C and pH 7.4.

| Sample | Exp. Time | $E_{corr}$ (mV $_{Ag/AgCl}$) | $i_{corr}$ ($\mu A/cm^2$) | $i_{pass}$ ($\mu A/cm^2$) |
|---|---|---|---|---|
| Ti-6Al-4V | 0 days | $105 \pm 14$ | $1.48 \pm 0.3$ | $41.7 \pm 4.1$ |
| | 90 days | $156 \pm 12$ | $0.29 \pm 0.08$ | $5.9 \pm 0.4$ |
| PEO | 0 days | $-115 \pm 4$ | $0.02 \pm 0.01$ | $1.5 \pm 0.2$ |
| | 90 days | $127 \pm 11$ | $0.01 \pm 0.001$ | $0.44 \pm 0.05$ |
| PEO-Ca/P | 0 days | $146 \pm 15$ | $0.006 \pm 0.0003$ | $0.14 \pm 0.008$ |
| | 90 days | $128 \pm 6$ | $0.0006 \pm 0.00005$ | $0.012 \pm 0.0007$ |

In good correlation with OCP measurements, the cathodic polarization branch was shifted to a lower potential for PEO sample 0 days. Literature data reported in some cases more negative $E_{corr}$ measurements for PEO-treated specimens than bulk Ti-6Al-4V [24,28], but in other cases more positive $E_{corr}$ has been obtained for Ca and P enriched PEO coating [25]. In our case, the Tafel curves show that the corrosion potential shifted to more positive values and to lower passive currents after the PEO treatment in all conditions, except for the PEO sample (0 days). Clearly, a more positive $E_{corr}$ indicates a higher chemical stability for the anodized layer compared to plain Ti-6Al-4V alloy. The lower corrosion potential observed initially (0 days) for the PEO specimen might be explained as a result the large amounts of defects and deep pores on the oxide layer but the exposition to AS led to an increase on the oxide layer chemical stability. The anodic polarization branch was more stable and the passive current densities were lower for PEO-coated samples than for non-coated samples. According to Table 5, the corrosion and passive current densities are approximately one order of magnitude lower than titanium alloy substrate for 0 day samples. This indicates again that the PEO coatings improve corrosion resistance of PM titanium before immersion testing.

The effect of the AS exposure on the anodic polarization behavior is clear for coated and non-coated samples. For non-modified samples the $E_{corr}$ increased and the $i_{corr}$ and $i_{pass}$ decreased after 90 days exposure due to the improvement in the resistance and in the oxide film thickness [31]. It was confirmed

that the $E_{corr}$ of PEO sample showed a strong shift in noble direction after the exposure and similar value to others samples was obtained. Also, the modified samples showed a relevant shift to left toward lower current densities after 90 days and the PEO-Ca/P sample showed the more important displacement. Accordingly, the lowest values of current densities and the highest values of corrosion potentials were found after PEO in electrolyte containing Ca and P ions.

Above results also confirm the effectiveness of using the PEO coating for obtaining the long-term corrosion resistant coatings for PM titanium alloys even much better than for conventional titanium alloys. The literature data, which describe behaviour of the PEO layers formed on titanium after long-term exposure, are scarce and contradictory. Kruppa et al [41] observed that resistance values for long-term exposures of PEO layers in SBF appear to be lower than those of non-oxidized titanium. The increase of the corrosion resistance of non-modified titanium can be attributed to the passive layer being rebuilt and thickened. In oxidized titanium, the corrosion resistance also increases during an exposure, but not as much as that of the non-oxidized titanium. The behavior of the PEO layers formed in a solution β-GP-Ca + calcium acetate differs from that observed in our samples: after long-term exposures, the corrosion resistance decreases [43]. However, similar behavior to our samples was observed on conventional titanium grade 2 [44]: the surface modification by PEO improves the corrosion resistance of titanium and it is not degraded after a long-term exposure in SBF, which was attributed to the increased thickness of the inner layer.

### 3.3. Cytocompatibility Tests

The evaluation of cell viability through the LIVE/DEAD assay showed that MC3T3-E1 preosteoblasts in contact with the different metallic surfaces were viable similarly to the positive control (Figure 7). Cells were also found to proliferate over time, as confirmed by the difference in the number of cells between day 1 and day 3.

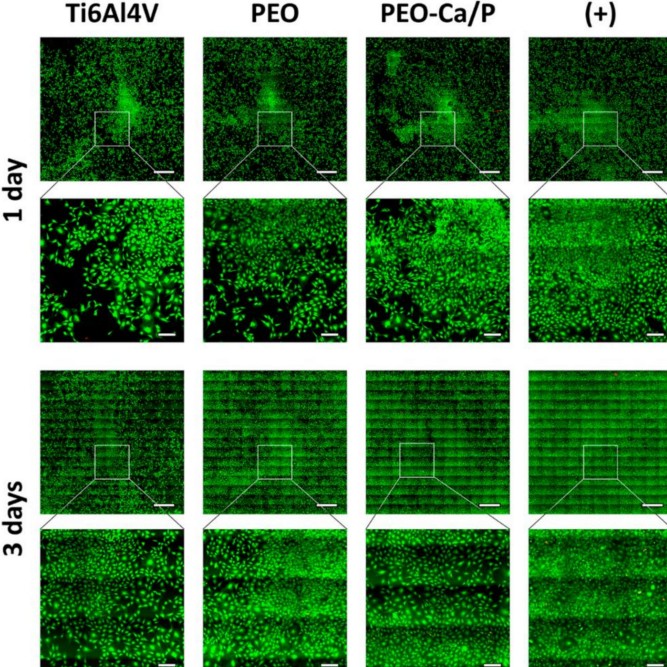

**Figure 7.** LIVE/DEAD assay of MC3T3-E1 preosteoblasts cultured in contact with different Ti surfaces: Ti6Al4V (uncoated Ti), PEO-coated Ti and PEO-Ca/P-coated Ti, in comparison with the positive control (+), after 1 and 3 days. Upper image for each timepoint shows a mosaic of almost the full well, while the lower image shows a detail of the mosaic. Scale bar for mosaic and detail images: 100 and 50 μm, respectively.

On the other hand, alamarBlue reduction results show that the presence of the uncoated and coated surfaces does not influence the metabolic activity of MC3T3-E1 cells, since no significant differences were found between the measurements obtained from cells in contact with the metallic surfaces and the positive control, at 1 and 3 days, Figure 8.

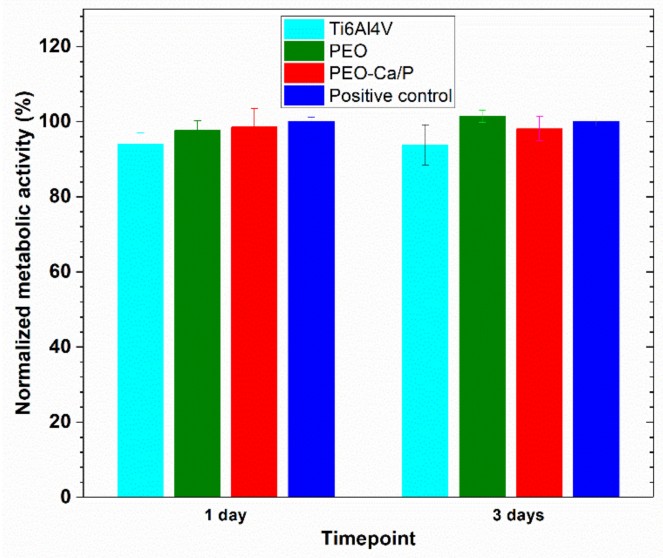

**Figure 8.** Metabolic activity of MC3T3-E1 cells in the presence of uncoated (Ti6Al4V) and coated (PEO and PEO-Ca/P) surfaces normalized to the positive control (considered as 100% metabolic activity), after 1 and 3 days of culture. No significant differences were found between the groups at each timepoint.

The results indicate that as it is known the Ti-4V-6Al alloy is one of the most cytocompatible biomaterials and that the surfaces covered with either PEO or PEO-Ca/P coatings are also cytocompatible and can be used in the generation of titanium implants suitable for biomedical applications, such as bone tissue engineering. Moreover, this cytocompatibility is also in good agreement with other studies that showed similar results with comparable wrought and cast titanium and titanium alloys [45,46].

## 4. Conclusions

PEO of Ti-6Al-4V alloy obtained by PM was studied in two electrolytes with different Ca and P contents. Homogeneous and adherent film with anatasa and rutile phases has been observed in these baths.

Electrochemical testing of coated and uncoated PM Ti-6Al-4V alloy in AS solution revealed that the PEO coatings showed not only an enhanced passivation behaviour without evidence of localized corrosion phenomena, but also a noble corrosion potential and a lower current density in the anodic branch of the polarization curve. Corrosion testing confirmed that the PEO-Ca/P coating shows the best corrosion performance even at long immersion times (90 days). The effect is better than for wrought and cast PEO coated materials.

In addition, preosteoblast cells seeded in the presence of the surfaces covered with either PEO or PEO-Ca/P coatings were highly viable results, even showing an increase in cell number over time as a consequence of cell proliferation, similarly to the positive control. This result, in combination with the metabolic activity measurements prove the cytocompatibility of the coatings. Therefore, they can be used in the generation of titanium implants suitable for biomedical applications.

Following PEO treatments, the performance of PM substrates has been improved, making them very attractive candidates for use in the generation of titanium implants suitable for biomedical applications. In future work, these coatings will be used for PM materials with a higher degree of porosity.

**Author Contributions:** The manuscript was written through contributions of all authors. All authors have given approval to the final version of the manuscript. C.G.-C. Conceptualization; Data curation; Methodology; Formal analysis; Investigation; original draft; Writing—review & editing. M.L.R.-M.: Project administration; Original draft; Funding acquisition. V.A.B.: Data curation and Formal analysis of materials; Methodology; Original draft. B.R.: Data curation and Formal analysis of coatings; Investigation; Original draft. A.I.F.: Data curation and Formal analysis of cytocompatibility tests; Investigation; Original draft. J.C.R.C.: Resources; Methodology. F.M.-P. Conceptualization.

**Funding:** Financial support by Ministry of Education and Science (Plan Nacional: RTI2018-097990-B-I00) and the Junta de Castilla y Leon (VA275P18) and ((VA044G19 is gratefully acknowledged. The authors are grateful for the funding from the Spanish Government (MAT2016-78903-R), Junta de Castilla y León (VA317P18), Interreg VA España Portugal POCTEP (0624_2IQBIONEURO_6_E) and Centro en Red de Medicina Regenerativa y Terapia Celular de Castilla y León.

**Conflicts of Interest:** The authors declare no conflict of interest.

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
