# Peer review of "Application of Plasma Electrolytic Oxidation Coating on Powder Metallurgy Ti-6Al-4V for Dental Implants"

_metals, doi:10.3390/met10091167_

Round 1
Reviewer 1 Report
This manuscript is generally well written. Thus, serious revision will not be needed. However, for the reader, I would like to comment some minor errors.
- make some space between panels (e.g., fig. 2)
- Revise the subscripts correctly. (e.g., Ecorr > Ecorr, TiO2 > TiO2) throughout the manuscript.
- Provide consistent number of significant digits in Table 5.
Author Response
This manuscript is generally well written. Thus, serious revision will not be needed. However, for the reader, I would like to comment some minor errors.
- make some space between panels (e.g., fig. 2)
- Spaces have already been made between the panels
- the subscripts correctly. (e.g., Ecorr > Ecorr, TiO2 > TiO2) throughout the manuscript.
- Sorry, when putting the manuscript into the format the sub-indexes were removed. The sub-indexes have been reviewed and corrected
- Provide consistent number of significant digits in Table 5.
-
- Some data in Table 5 have been approximated

Reviewer 2 Report
The manuscript submitted to Metals journal intends to study the role of PEO coatings on PM Ti-6Al-4V obtained by PM. In this study, two different electrolyte solutions, were considered. The corrosion performance in AS simulated body fluids was electrochemically evaluated by OCP, and potentiodynamic polarization curves. The authors also discussed the correlation between microstructure and corrosion resistance.
The manuscript is generally well written. However, before publication some major concerns should be addressed, namely:
- The introductions need to be improved since as presented it does not reflect the actual state-of-the-art.
- How many samples were studied? Duplicates, triplicates? This should be clearly indicated.
- I also propose shortening the conclusion section.

Author Response
The manuscript submitted to Metals journal intends to study the role of PEO coatings on PM Ti-6Al-4V obtained by PM. In this study, two different electrolyte solutions, were considered. The corrosion performance in AS simulated body fluids was electrochemically evaluated by OCP, and potentiodynamic polarization curves. The authors also discussed the correlation between microstructure and corrosion resistance.
The manuscript is generally well written. However, before publication some major concerns should be addressed, namely:
- The introductions need to be improved since as presented it does not reflect the actual state-of-the-art.
- The introduction has been implemented and the current references have been introduced.
- How many samples were studied? Duplicates, triplicates? This should be clearly indicated.
-
- The tests have been carried out in triplicate. This has been indicated in the section on experimental methods
- I also propose shortening the conclusion section.
-
- The conclusion section has been shorted

Reviewer 3 Report
Manuscript: metals-887919
Title: Application of plasma electrolytic oxidation coating on PM Ti-6Al-4V for dental implants
Report:
The authors applied two plasma electrolytic oxidation processes on sintered Ti6Al4V samples. The comparative study included X-ray diffraction, X-ray fluorescence, electron microscopy, corrosion in artificial saliva and cytocompatibility tests. The results indicate that the PEO process formed a 16 um thick layer on the Ti6Al4V substrate, enriched with Ca and P. This layer increased the corrosion resistance of the PM Ti6Al4V specimens, particularly after the surface defects (cracks and pores) were sealed by corrosion products. In addition, the cytocompatibility tests revealed a similar response to the positive reference.
The article is interesting and novel. I consider it would be acceptable for publication after minor corrections. Below is a list of suggestions for the authors:
- Abstract: The authors refer to the inherent porosity of the powder metallurgy components as detrimental for the corrosion resistance and later refer to the “proper porosity index” with regards to osseointegration and biocompatibility. These concepts are not fully explained in the introduction and in the discussion. Please expand the introduction and/or discussion to explain these two concepts related with the porosity of the PM Ti6Al4V or the PEO coating and the expected effect on corrosion and stress-shielding.
- Materials: The authors used a mixture of elemental powders to produce the samples by powder metallurgy. Please indicate the average particle size and how the powders were mixed before the sintering process to avoid chemical segregation.
- Methods: Please describe the sample preparation of the Ti6Al4V samples prior to the PEO process. Where the samples ground or cleaned in any way? Please include this in the methods.
- Methods: The equipment used to measure the surface roughness of the samples is not described. Please describe this equipment and the conditions used for the measurements.
- Results and discussion: The chemical analyses of the samples revealed the presence of iron. Was iron present in the elemental powders or is it introduced in the samples as contamination during the sintering process? Please clarify.
- Results and discussion: The authors indicate that surface roughness is important for cell adhesion and proliferation, but no information is provided with regards to the target value. Please clarify (see also comment 1).
- Results and discussion: The Keronite Electrolyte 105 1003-01 for titanium is said to contain 0.1–1 % sodium fluoride (NaF), but no F is shown in the maps. Is this because of detection limits of the technique or the absence of F in the coatings? Please clarify.
- Results and discussion, Figure 4: Please consider removing the background from the diffractogram.
- Results and discussion, lines 278 to 280: Please review and delete. These lines seem to be comments between the authors.
- Results and discussion: The description of the corrosion curves indicates that “the current density was negative” in the first domain (line 316). Please review and clarify this.
- Results and discussion: The authors refer to conflicting evidence in the literature with regards to the Ecorr measurements on the PEO layers. Please consider expanding this to indicate your interpretation of these differences in the reported data.
- Conclusions: Please review the last two lines (391 and 392) and delete if appropriate. These lines seem to be comments between the authors.
Author Response
Report:
The authors applied two plasma electrolytic oxidation processes on sintered Ti6Al4V samples. The comparative study included X-ray diffraction, X-ray fluorescence, electron microscopy, corrosion in artificial saliva and cytocompatibility tests. The results indicate that the PEO process formed a 16 um thick layer on the Ti6Al4V substrate, enriched with Ca and P. This layer increased the corrosion resistance of the PM Ti6Al4V specimens, particularly after the surface defects (cracks and pores) were sealed by corrosion products. In addition, the cytocompatibility tests revealed a similar response to the positive reference.
The article is interesting and novel. I consider it would be acceptable for publication after minor corrections. Below is a list of suggestions for the authors:
Abstract:
- The authors refer to the inherent porosity of the powder metallurgy components as detrimental for the corrosion resistance and later refer to the “proper porosity index” with regards to osseointegration and biocompatibility. These concepts are not fully explained in the introduction and in the discussion. Please expand the introduction and/or discussion to explain these two concepts related with the porosity of the PM Ti6Al4V or the PEO coating and the expected effect on corrosion and stress-shielding.
- These aspects have been further discussed the introduction.
Materials:
- The authors used a mixture of elemental powders to produce the samples by powder metallurgy. Please indicate the average particle size and how the powders were mixed before the sintering process to avoid chemical segregation.
- Such information has been included in materials subsection.
Methods:
- Please describe the sample preparation of the Ti6Al4V samples prior to the PEO process. Where the samples ground or cleaned in any way? Please include this in the methods.
- This is included in the experimental methods.
- The equipment used to measure the surface roughness of the samples is not described. Please describe this equipment and the conditions used for the measurements.
- This has been described in the experimental methods section.
Results and discussion:
- The chemical analyses of the samples revealed the presence of iron. Was iron present in the elemental powders or is it introduced in the samples as contamination during the sintering process? Please clarify.
- In the raw material there is a 0.2% Fe, which is normal in any titanium powder, and in this case in Ti-6Al-4V. But in table 2, a chemical analysis has been done on the different constituents, that show that the iron becomes completely part of the beta phase (≈1%). As you can see, in the alpha phase matrix no iron is observed, but in its grain boundary -which corresponds to the beta phase- Fe appears back on a 1%. In addition, vanadium is accumulated with contents around 15%, whereas the total of the alloy only has just a 4%.
- The authors indicate that surface roughness is important for cell adhesion and proliferation, but no information is provided with regards to the target value. Please clarify (see also comment 1).
- This has already been clarified.
- Results and discussion: The Keronite Electrolyte 105 1003-01 for titanium is said to contain 0.1–1 % sodium fluoride (NaF), but no F is shown in the maps. Is this because of detection limits of the technique or the absence of F in the coatings? Please clarify.
- The presence of F in the commercial electrolyte is very low. Accordingly, it is expected that the amount of fluoride present in the coating structure will be so small that it will be difficult to detect with SEM-EDS.
- Figure 4: Please consider removing the background from the diffractogram.
- The baseline has been corrected, but the presence of amorphous phase increases the intensities in the region around 2θ= 20-40.
- lines 278 to 280: Please review and delete. These lines seem to be comments between the authors.
- I'm sorry it's already been removed
- The description of the corrosion curves indicates that “the current density was negative” in the first domain (line 316). Please review and clarify this.
- It has been clarified in the text.
- The authors refer to conflicting evidence in the literature with regards to the Ecorr measurements on the PEO layers. Please consider expanding this to indicate your interpretation of these differences in the reported data.
- It has been included in the discussion.
Conclusions:
- Please review the last two lines (391 and 392) and delete if appropriate. These lines seem to be comments between the authors.
- I'm sorry it's already been removed

Round 2
Reviewer 2 Report
I support publication since all the issues raised were well addressed.